# Impact of Chronic Obstructive Pulmonary Disease on Incidence, Microbiology and Outcome of Ventilator-Associated Lower Respiratory Tract Infections

**DOI:** 10.3390/microorganisms8020165

**Published:** 2020-01-23

**Authors:** Anahita Rouzé, Pauline Boddaert, Ignacio Martin-Loeches, Pedro Povoa, Alejandro Rodriguez, Nassima Ramdane, Jorge Salluh, Marion Houard, Saad Nseir

**Affiliations:** 1Centre Hospitalier Universitaire Lille, Critical Care Center, F-59000 Lille, France; anahita.rouze@chru-lille.fr (A.R.); pauline.boddaert@vhru-lille.fr (P.B.); marion.houard@chru-lille.fr (M.H.); 2Medicine Faculty, University of Lille, F-59000 Lille, France; 3Department of Intensive Care Medicine, Multidisciplinary Intensive Care Research Organization (MICRO), St. James’s Hospital, St. James Street, Dublin 8, D08 NHY1 Dublin, Ireland; drmartinloeches@gmail.com; 4Hospital Clinic, IDIBAPS, Universidad de Barcelona, Ciberes, 08036 Barcelona, Spain; 5Nova Medical School, New University of Lisbon, 1099-085 Lisbon, Portugal; 6ICU, Hospital Universitari de Tarragona Joan XXIII, 43005 Tarragona, Spain; ahr1161@yahoo.es; 7Centre Hospitalier Universitaire Lille, University of Lille, EA 2694—Santé Publique, Epidémiologie et Qualité des Soins, Département de Biostatistiques, F-59000 Lille, France; nassima.ramdane@chru-lille.fr; 8Institute for Research and Education, D’Or, Rio de Janeiro 22281-100, Brazil; jorgesalluh@gmail.com

**Keywords:** chronic obstructive pulmonary disease, ventilator-associated, lower respiratory tract infections, pneumonia, tracheobronchitis, mechanical ventilation, intensive care

## Abstract

Objectives: To determine the impact of chronic obstructive pulmonary disease (COPD) on incidence, microbiology, and outcomes of ventilator-associated lower respiratory tract infections (VA-LRTI). Methods: Planned ancillary analysis of TAVeM study, including 2960 consecutive adult patients who received invasive mechanical ventilation (MV) > 48 h. COPD patients (*n* = 494) were compared to non-COPD patients (*n* = 2466). The diagnosis of ventilator-associated tracheobronchitis (VAT) and ventilator-associated pneumonia (VAP) was based on clinical, radiological and quantitative microbiological criteria. Results: No significant difference was found in VAP (12% versus 13%, *p* = 0.931), or VAT incidence (13% versus 10%, *p* = 0.093) between COPD and non-COPD patients. Among patients with VA-LRTI, *Escherichia coli* and *Stenotrophomonas maltophilia* were significantly more frequent in COPD patients as compared with non-COPD patients. However, COPD had no significant impact on multidrug-resistant bacteria incidence. Appropriate antibiotic treatment was not significantly associated with progression from VAT to VAP among COPD patients who developed VAT, unlike non-COPD patients. Among COPD patients, patients who developed VAT or VAP had significantly longer MV duration (17 days (9–30) or 15 (8–27) versus 7 (4–12), *p* < 0.001) and intensive care unit (ICU) length of stay (24 (17–39) or 21 (14–40) versus 12 (8–19), *p* < 0.001) than patients without VA-LRTI. ICU mortality was also higher in COPD patients who developed VAP (44%), but not VAT(38%), as compared to no VA-LRTI (26%, *p* = 0.006). These worse outcomes associated with VA-LRTI were similar among non-COPD patients. Conclusions: COPD had no significant impact on incidence or outcomes of patients who developed VAP or VAT.

## 1. Introduction

Chronic obstructive pulmonary disease (COPD) is characterized by persistent airflow limitation, associated with an enhanced chronic inflammatory response in the airways and the lung to noxious particles or gases [1]. According to the WHO, over 65 million people have moderate to severe COPD worldwide. With an increasing prevalence, COPD is thought to become the third cause of death in the world by 2030 [2]. In a large Austrian cohort including 194,453 critically ill patients [3], COPD was present in 8.6% of all patients, including one third admitted for acute respiratory failure due to COPD and two thirds with only comorbid COPD. Although use of noninvasive ventilation in acute exacerbations of COPD has substantially reduced the need for intubation, a large proportion of COPD patients still require invasive mechanical ventilation (MV). The largest prospective international cohorts of mechanically ventilated patients reported 10% of critically ill patients receiving MV because of acute exacerbation of COPD in 1998, and still 6% in 2010 [4].

Ventilator-associated lower respiratory tract infections (VA-LRTI), including both ventilator-associated tracheobronchitis (VAT) and ventilator-associated pneumonia (VAP), are the most common healthcare-associated infections among critically ill patients receiving MV. Approximately, one out of four patients intubated for more than 48 h develop VA-LRTI [5]. Intubated COPD patients may be at increased risk for VA-LRTI. Specific risk factors in this population include: (1) prolonged duration of invasive MV, linked with skeletal and diaphragmatic muscle weakness; (2) high incidence of microaspiration, due to gastro-esophageal reflux and altered interaction between breathing and deglutition, resulting in high tracheobronchial bacterial colonization, also favored by defective mucociliary clearance; and (3) altered general host defense mechanisms [6].

Several studies have identified COPD as independently associated to VAP occurrence [7,8], but more recent data suggest that COPD patients may not be at higher risk for VAP [9]. However, VAP is associated with prolonged MV and intensive care unit (ICU) stay, and higher mortality among critically ill patients [5,10], including COPD population [9,11], although mortality attributable to VAP is still a matter for debate [12].

Further, VAT has been proposed as an intermediate stage between tracheobronchial colonization and VAP [13,14]. Several studies have already reported that VAT could progress to VAP, and was associated with longer duration of MV and ICU stay [15,16,17,18]. Recently, our group have confirmed that VAT was a separate entity, responsible itself for increased duration of MV and ICU stay in a large cohort of medical and surgical patients [5]. VAT was not associated with increased mortality rates, but transition from VAT to VAP was a risk factor for mortality and appropriate antibiotic treatment decreased both subsequent pneumonia rate and mortality.

However, to our knowledge, no study to date has specifically evaluated VAT among COPD population. One could hypothesize a higher incidence of VAT, as well as worse outcomes, in this particular population. Additionally, as previously mentioned, conflicting epidemiological data exist regarding the impact of COPD on the incidence of VAP. Therefore, we conducted this study to determine the impact of COPD on the incidence, microbiology, and outcomes of patients with VA-LRTI.

## 2. Patients and Methods

### 2.1. Study Design and Population 

This study is a planned ancillary analysis of TAVeM study [5]. TAVeM is a large prospective multinational observational study conducted in 114 ICUs in Europe (Spain, France, Portugal) and South America (Brazil, Argentina, Ecuador, Bolivia and Colombia), which consecutively included, between September 2013 and July 2014, 2960 patients older than 18 years, who received invasive MV for more than 48 h, and prospectively collected data regarding VA-LRTI, defined as VAT and VAP. Readmitted patients and patients who had been previously tracheostomised were not included. Participating centers either received ethics approval from their institutions or ethics approval was waived (institutional review board number 2013515). Informed consent was waived because of the observational nature of the study.

### 2.2. Data Collection

Patients were prospectively followed up for VA-LRTI diagnosis and outcome data until death or discharge from hospital [5]. Patient demographic characteristics, severity scores, comorbidities, primary diagnoses, prior antibiotic exposure (during the three months preceding ICU admission) were recorded at baseline for all patients. Further, data about clinical, biological and radiological diagnostic criteria for VA-LRTI, microbiological diagnostic procedures, bacteriological findings, degree of severity on the onset of infection, antibiotic use and clinical outcomes were obtained.

### 2.3. Definitions

#### 2.3.1. Chronic Obstructive Pulmonary Disease

Patients were considered as having COPD if exhibiting documented medical history, namely pulmonary function tests before admission, but also compatible clinical and/or radiological criteria, such as smoking exposure, history of acute exacerbations, hyperinflation or emphysema on chest X-ray, expiratory flow limitation during controlled MV [1].

#### 2.3.2. Ventilator-Associated Lower Respiratory Tract Infection

The diagnosis of VA-LRTI was based on the presence of at least two of the following criteria [5]: body temperature of more than 38.5 °C or less than 36.5 °C, leucocyte count greater than 12,000 cells per μL or less than 4000 cells per μL, and purulent secretions. Additionally, all episodes of infection needed microbiological confirmation, with the isolation in the endotracheal aspirate of at least 10^5^ colony-forming units (CFU) per mL, or in bronchoalveolar lavage of at least 10^4^ CFU per mL, to be included in the final analysis. 

VAT was defined with the above-mentioned criteria with no radiographical signs of new pneumonia. VAP was defined by the presence of new or progressive infiltrates on chest radiograph. 

Only first episodes of VAT and VAP were analyzed. VAP was defined as occurring subsequently to VAT if it was diagnosed in the 96 h period after diagnosis of VAT and if the same microorganism caused both infections.

#### 2.3.3. Antibiotic Treatment and Microbiological Data

Antibiotic was considered appropriate if it matched the in vitro susceptibility of the pathogen causing VA-LRTI [19]. Microbiological identification and susceptibility tests were performed using standard methods. Multidrug resistant (MDR) isolates were defined as acquired non-susceptibility to at least one agent in three or more antimicrobial categories [20].

### 2.4. Outcomes 

The primary aim of our study was to determine the impact of COPD on the incidence of VA-LRTI, including VAT and VAP. 

Our secondary objectives were to determine the impact of COPD on the microbiology and outcomes (duration of MV, length of stay in the ICU, in the hospital, and ICU mortality) of VA-LRTI. We also studied diagnostic data, antibiotic treatment data, and the effect of antibiotic treatment on progression from VAT to VAP in COPD as compared to non-COPD patients.

### 2.5. Statistical Analysis 

Categorical variables were expressed as numbers (percentages) and compared using Chi-square test or Fisher’s exact test, as appropriate. Normality of distribution of continuous variables was checked graphically and by using the Shapiro–Wilk test. As all continuous variables were skewed, they were presented as medians (interquartile ranges), and compared using Mann–Whitney *U* test or Kruskal–Wallis test. Adjusted analyses on age and gender were performed using non-parametric analysis of covariance for continuous variables and logistic regression for categorical variables. Univariate and multivariate analyses to determine factors associated with ICU mortality in patients with ventilator associated lower respiratory tract infection were also performed. All variables with a *p*-value < 0.1 in univariate analysis were included in a logistic regression model using a stepwise backward elimination. All statistical tests were two-tailed, and *p* values < 0.05 were considered statistically significant. The Statistics Department of Lille University Hospital performed all data analyses using SAS software package, release 9.4 (SAS Institute, Cary, NC, USA).

## 3. Results

### 3.1. Baseline Patient Characteristics

Among the 2960 included patients, 494 (17%) had COPD. Age, percentage of men, percentage of patients with chronic diseases such as diabetes mellitus, chronic respiratory failure, chronic kidney disease, and alcohol abuse, were significantly higher in COPD patients compared with non-COPD patients. COPD patients were significantly more frequently admitted to the ICU for medical reasons, with a higher percentage of pneumonia as the main cause of admission, compared with non-COPD patients (Table 1). 

Prior antibiotic use was significantly more frequent in COPD patients compared with non-COPD patients who developed VA-LRTI (95 of 121 (78.5%) versus 347 of 530 (65.5%), *p* = 0.0056). 

### 3.2. Incidence of Ventilator-Associated Lower Respiratory Tract Infections

No significant difference was found in the incidence of VAP (12% versus 13%, *p* = 0.931), or VAT (13% versus 10%, *p* = 0.093) between COPD and non-COPD patients (Figure 1). No significant difference was found in the rate of transition from VAT to VAP (17% versus 11% of VAT episodes, *p* = 0.172) between COPD and non-COPD patients.

### 3.3. Characteristics of Ventilator-Associated Lower Respiratory Tract Infections

#### 3.3.1. Diagnostic Data

On the day of VA-LRTI diagnosis, SOFA score and PCT were not significantly different in COPD patients who developed VAP as compared to patients who developed VAT, unlike in non-COPD patients (Table 2). However, clinical pulmonary infection score (CPIS) was significantly higher in VAP than in VAT patients, both in COPD and non-COPD population.

In both COPD and non-COPD patients, bronchoalveolar lavage was significantly more frequently used to diagnose VAP than VAT. Nevertheless, significantly higher proportion of COPD patients had invasive microbiological diagnostic tests, i.e., bronchoscopy or bronchoalveolar lavage, than did non-COPD patients, in both VAT and VAP groups (Table 2).

#### 3.3.2. Microbiological Data

Among patients with VA-LRTI, *Escherichia coli* and *Stenotrophomonas maltophilia* were significantly more frequent in COPD patients as compared with non-COPD patients (Table 3). The rate of MDR bacteria was not significantly different between COPD and non-COPD patients.

#### 3.3.3. Impact of Antibiotic Treatment on Transition from VAT to VAP 

In COPD patients, no significant difference was found in rate of antibiotic treatment, appropriate antibiotic treatment, or duration of antibiotic treatment between patients with transition from VAT to VAP, as compared with those with no transition from VAT to VAP (Table 4). 

### 3.4. Outcomes of Ventilator-Associated Lower Respiratory Tract Infections

In COPD patients, MV, ICU and hospital length of stay were significantly different between patients with VAT, VAP, or no VA-LRTI, as in non-COPD patients. ICU mortality was also significantly different in these three groups of patients in both COPD and non-COPD populations: VAP was associated with a higher mortality compared to VAT and no-VALRTI among non-COPD patients, and only compared to no-VALRTI among COPD patients. COPD had no significant impact on duration of MV, length of ICU or hospital stay, or mortality in patients with VA-LRTI. In patients without VA-LRTI, COPD patients exhibited significantly lower ICU mortality, as compared to non-COPD patients (Table 5). Factors associated with ICU mortality in patients with VA-LRTI in univariate and multivariate analyses are shown in Appendix A.

## 4. Discussion

The main results of our study are as follows: (1) the incidence of VAP and VAT was not significantly different between COPD and non-COPD patients. (2) Among patients with VA-LRTI, *E. coli* and *S. maltophilia* were significantly more frequent in COPD patients as compared with non-COPD patients. The rate of MDR bacteria was not significantly different between the two groups. (3) Among COPD patients, those with VAT or VAP had significantly longer MV duration, ICU and hospital length of stay, as compared to patients without VA-LRTI. ICU mortality was also higher among COPD patients who developed VAP as compared to patients without VA-LRTI. These worse outcomes associated with VA-LRTI were similarly found among non-COPD patients. (4) Appropriate antibiotic treatment was not significantly associated with progression from VAT to VAP among COPD patients who developed VAT, unlike non-COPD patients.

To the best of our knowledge, our study is the first to evaluate the relationship between COPD and VA-LRTI, specifically including VAT. It benefits from several strengths of TAVeM study, which constitutes the largest prospective multicenter international cohort of VA-LRTI, with strict diagnostic criteria, obtaining data from consecutive patients, and generating robust and reproducible results [5].

Among intubated COPD patients, VAP incidence widely ranges, from 6 to 34% [7,8,9,11], possibly because of different VAP diagnostic criteria used in these studies. The relatively low rate of VAP among COPD patients, together with the lack of significant difference with non-COPD patients, may be due to the significantly higher previous exposure to antibiotics in COPD patients. Previous studies reported that antibiotic treatment could be protective against VAP [21]. However, antibiotic treatment is well-known risk factor for late-onset ICU-acquired infections, and MDR bacteria emergence [22]. Our results contradict several earlier studies that identified COPD as an independent risk factor for VAP [7,8], but are in line with data from a recent retrospective analysis of a large prospective observational study conducted in 27 European ICUs and including 2082 intubated patients, which reported a similar 20% incidence of VAP among COPD and non COPD patients [9]. 

In addition, differentiating VAT and VAP episodes in COPD patients can be challenging, as the interpretation of chest X-ray to identify new or progressive infiltrates, may be particularly tricky in this population with baseline abnormal radiographs. This might have resulted in a misclassification of VA-LRTI episodes. Using CT scan or lung ultrasound to diagnose pneumonia may be a future option to effectively differentiate VAT and VAP in COPD patients, as some authors have recently proposed for community-acquired pneumonia [23,24]. However, baseline CT would be needed, and indications should be carefully evaluated, as transport of critically ill patients to the radiology department was repeatedly reported to be associated with adverse events [25].

Little but consistent data are available on the microbiology of VAP in COPD patients [9,11,26]. Gram-negative bacilli are the most frequently isolated microorganisms in this population, with a high proportion of *Pseudomonas aeruginosa*, around one third of responsible bacteria, followed by *Acinetobacter baumannii* and *Staphylococcus aureus*. Koulenti et al. recently reported a higher prevalence of *P. aeruginosa* in VAP among COPD patients, as compared to non COPD patients (29 versus 19%, *p* = 0.04) [9]. The high proportion of MDR bacteria in COPD patients with VAP could be explained by the frequent antimicrobial treatments and hospitalizations in this population [27]. Our results showed significantly higher rates of *S. maltophilia* and *E. coli* in COPD patients as compared with non-COPD patients. However, no significant difference was found in MDR bacteria, or *P. aeruginosa* rates between COPD, and non-COPD patients. COPD has been previously reported as an independent risk factor for *S. maltophilia* ICU-acquired infections [28]. The absence of significant difference in MDR bacteria rate between COPD and non-COPD patients could be explained by high rates of MDR bacteria in both COPD and non-COPD population, and different local epidemiology. 

Our results suggest that COPD itself is not associated with negative impact on outcome of VAP or VAT patients, as same worse outcomes were observed among COPD and non-COPD patients in our cohort, as compared to patients without VA-LRTI. Our results are in line with the recent large European cohort of intubated critically ill patients, including 397 COPD patients, which reported a 17% significant increase of ICU mortality in COPD patients who developed VAP, as compared to COPD patients without VAP, and identified the occurrence of VAP as an independent predictor of mortality among COPD patients (odds ratio 2.28, 95% confidence interval 1.35–3.87) [9]. However, a prospective observational study including 215 patients with VAP, observed a significantly higher ICU mortality in COPD patients compared with non COPD patients (60 versus 43%, *p* = 0.027) and identified COPD as an independent risk factor for ICU mortality in multivariate analysis (odds ratio 2.58, 95% confidence interval 1.34–5) [26]. At least two potential explanations could be given for the absence of significant impact of COPD on outcomes of VAP patients in our study. First, our study might have been underpowered to detect such an effect. Second, the definition used for COPD was probably not specific, as lung function testing results were not available.

Besides, COPD has been considered a risk factor of death in ICU. A quite recent prospective study including 235 ventilated patients without evidence of respiratory infection showed that COPD was independently associated with mortality (hazard ratio 2.1 [1.10–3.94]) [29]. Interestingly, our data suggest the opposite in the subgroup of patients without VA-LRTI, in which COPD was associated with significantly lower ICU mortality. 

The rate of progression from VAT to VAP was similar in COPD and non-COPD patients in our study, suggesting that COPD is not a risk factor for progression from VAT to VAP. Antibiotic treatment, either appropriate or not, did not significantly reduce the progression from VAT to VAP among COPD patients. This result could be explained by the small number of COPD patients with transition from VAT to VAP.

Our study has several limitations. First, this was a planned analysis of a prospective database not initially designed to evaluate the relationship between COPD and VA-LRTI. The presence of COPD was recorded by the attending physician, based on pulmonary function tests before ICU admission, or compatible clinical and/or radiological criteria. Furthermore, data regarding the severity of COPD, such as GOLD stages, long term oxygen therapy, noninvasive ventilation at home or previous corticosteroid use were not recorded, and results might have been different if specific severity subgroups have been analyzed.

## 5. Conclusions

Our study suggests that COPD does not impact the incidence or the outcome of VA-LRTI. However, higher rates of *S. maltophilia*, and *E. coli* were found in COPD patients with VA-LRTI, as compared with non COPD patients.

## Figures and Tables

**Figure 1 microorganisms-08-00165-f001:**
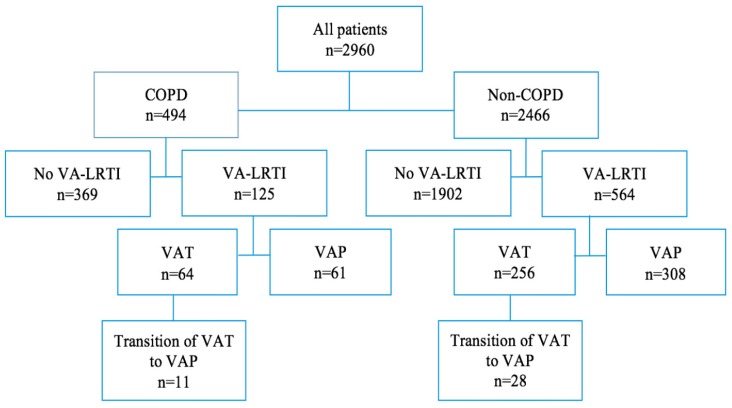
Study flowchart. Data are presented as number of patients. Only first episodes of VA-LRTI were taken into account. COPD, Chronic Obstructive Pulmonary Disease; VA-LRTI, ventilator-associated lower respiratory tract infection; VAP, ventilator-associated pneumonia; VAT, ventilator-associated tracheobronchitis.

**Table 1 microorganisms-08-00165-t001:** Patient characteristics at intensive care unit (ICU) admission in chronic obstructive pulmonary disease (COPD) and non-COPD patients.

	COPD *n* = 494	No COPD *n* = 2466	*p*
Age, years	68 (60–76)	63 (49–74)	<0.001
Male gender	355 (72)	1494 (61)	<0.001
Severity scores			
SAPS II	50 (39–62)	50 (37–63)	0.27
SOFA score	8 (5–10)	8 (5–11)	0.29
Comorbidities			
Diabetes mellitus	126 (26)	442 (18)	<0.001
Chronic respiratory failure	187 (38)	99 (4)	<0.001
Chronic heart failure	36 (7)	176 (7)	0.91
Chronic kidney disease	76 (15)	217 (9)	<0.001
Cirrhosis	25 (5)	152 (6)	0.35
Immunosuppression	113 (23)	549 (22)	0.77
Alcohol abuse	84 (17)	274 (11)	<0.001
Intravenous drug abuse	10 (2)	43 (2)	0.67
Category of admission			<0.001
Medical	392 (79)	1496 (61)	
Surgical	68 (14)	476 (19)	
Trauma	34 (7)	494 (20)	
Main causes for ICU admission			
Shock	33 (7)	238 (10)	0.037
Sepsis	54 (11)	284 (12)	0.71
Acute respiratory distress syndrome	23 (5)	124 (5)	0.73
Pneumonia	110 (22)	308 (12)	<0.001
Aspiration	5 (1)	44 (2)	0.22
Congestive heart failure	20 (4)	76 (3)	0.27
Myocardial infarction	12 (2)	57 (2)	0.87
Arrhythmia	6 (1)	28 (1)	0.88
Coma	26 (5)	248 (10)	<0.001
Stroke	6 (1)	115 (5)	<0.001
Seizure	5 (1)	66 (3)	0.027
Brain aneurysm	5 (1)	37 (2)	0.40
Traumatic brain injury	6 (1)	153 (6)	<0.001
Acute renal failure	9 (2)	36 (1)	0.55

Data are presented as number (%) or median (interquartile range). COPD, chronic obstructive pulmonary disease; ICU, intensive care unit; SAPS, simplified acute physiology score; SOFA, sequential organ failure assessment. Some patients had more than one cause for ICU admission.

**Table 2 microorganisms-08-00165-t002:** Biological criteria and microbiological diagnostic procedures used in ventilator-associated lower respiratory tract infection diagnosis in COPD and non-COPD patients.

	COPD *n* = 125			No COPD N = 564		
	VAT *n* = 64	VAP *n* = 61	*p*	VAT *n* = 256	VAP *n* = 308	*p*
CRP	69 (9–201)	129 (17–230)	0.374	126 (51–211)	120 (30–239)	0.78
PCT	0.1 (0.0–1.7)	0.8 (0.0–3.8)	0.219	0.1 (0.0–1.0)	0.4 (0.0–5.0)	<0.001
CPIS	5 (3–6)	6 (5–8)	<0.001	4 (3–6)	7 (5–8)	<0.001
Endotracheal aspirate	52 (81)	41 (67)	0.072	226 (88)	212 (69)	<0.001
Bronchoscopy	7 (11) *	15 (25) *	0.045	10 (4)	41 (13)	<0.001
Bronchoalveolar lavage	6 (9) **	15 (25) **	0.023	6 (2)	44 (14)	<0.001
Mini bronchoalveolar lavage	5 (8)	4 (7)	0.786	11 (4)	41 (13)	<0.001
Blind protected specimen brush	8 (13)	12 (20)	0.274	24 (9)	39 (13)	0.22

Data are presented as number (%) or median (interquartile range). COPD, chronic obstructive pulmonary disease; VAP, ventilator-associated pneumonia; VAT, ventilator-associated tracheobronchitis; CRP, C-reactive protein; PCT, procalcitonin; CPIS, clinical pulmonary infection score. *p* values are for comparison between VAT and VAP groups. * *p* = 0.008 and ** *p* = 0.005 versus patients with no COPD.

**Table 3 microorganisms-08-00165-t003:** Microbiological findings in COPD and non-COPD patients with ventilator-associated lower respiratory tract infections.

	COPD *n* = 125	No COPD *n* = 564	*p*
Polymicrobial	33 (26)	132 (23)	0.48
Multidrug-resistant isolates	75 (60)	346 (62)	0.71
Gram-negative bacilli			
*Pseudomonas aeruginosa*	32 (26)	136 (24)	0.73
*Escherichia coli*	23 (18)	54 (10)	0.005
*Klebsiella pneumoniae*	15 (12)	86 (15)	0.35
*Enterobacter* spp.	15 (12)	66 (12)	0.93
*Stenotrophomonas maltophilia*	11 (9)	20 (4)	0.010
*Haemophilus influenzae*	7 (6)	50 (9)	0.23
*Proteus mirabilis*	7 (6)	22 (4)	0.39
*Serratia marcescens*	5 (4)	23 (4)	0.97
*Acinetobacter baumannii*	4 (3)	37 (7)	0.15
*Citrobacter freundii*	3 (2)	10 (2)	0.71
Gram-positive cocci			
MSSA	21 (17)	125 (22)	0.18
MRSA	4 (3)	12 (2)	0.51
*Streptococcus pneumoniae*	8 (6)	32 (6)	0.75

Data are presented as number (%). COPD, chronic obstructive pulmonary disease; MRSA, meticillin-resistant *Staphylococcus aureus;* MSSA, meticillin-sensitive *Staphylococcus aureus*.

**Table 4 microorganisms-08-00165-t004:** Antibiotic use in COPD and non-COPD patients with ventilator-associated tracheobronchitis.

	COPD *n* = 64			No COPD *n* = 256		
	VAT to VAP Progression *n* = 11	No VAT to VAP Progression *n* = 53	*p*	VAT to VAP Progression *n* = 28	No VAT to VAP Progression *n* = 228	*p*
Antibiotic treatment	10 (91)	50 (94)	0.539	21 (75)	213 (93)	0.005
Appropriate antibiotic treatment	6 (55)	42 (79)	0.124	13 (46)	189 (83)	<0.001
Length of antibiotic treatment, days	7 (3–8)	7 (4–10)	0.591	6 (4–10)	7 (4–10)	0.99

Data are presented as number (%) or median (interquartile range). COPD, chronic obstructive pulmonary disease; VAP, ventilator-associated pneumonia; VAT, ventilator-associated tracheobronchitis. *p* values are for comparison between VAT to VAP and no VAT to VAP groups.

**Table 5 microorganisms-08-00165-t005:** Clinical outcomes of study patients.

	COPD *n* = 494				No COPD *n* = 2466			
	VAT *n* = 64	VAP *n* = 61	No VA-LRTI *n* = 369	*p*	VAT *n* = 256	VAP *n* = 308	No VA-LRTI *n* = 1902	*p*
MV duration, days	17(9–30)	15(8–27)	7(4–12)	<0.001	13(8–21)	15(8–26)	7(4–13)	<0.001
ICU length of stay, days	24(17–39)	21(14–40)	12(8–19)	<0.001	20(14–31)	21(13–33)	12(8–19)	<0.001
Hospital length of stay, days	42(22–59)	30(18–56)	23(14–38)	<0.001	36(21–54)	31(20–54)	23(14–41)	<0.001
ICU mortality	24 (38) ^$^	27 (44)	95 (26) *	0.006	68 (27) ^£^	116 (38)	565 (30)	<0.001

Data are presented as number (%) or median (interquartile range). *P* values are adjusted on age and gender using non-parametric analysis of covariance for continuous variables and logistic regression for categorical variables. COPD, chronic obstructive pulmonary disease; ICU, intensive care unit; MV, mechanical ventilation; VA-LRTI, ventilator-associated lower respiratory tract infection; VAP, ventilator-associated pneumonia; VAT, ventilator-associated tracheobronchitis. *P* values are for comparisons between the three groups. ^$^
*p* > 0.999 versus VAP, *p* = 0.150 versus no VA-LRTI, ^£^
*p* = 0.002 versus VAP, *p* > 0.999 versus no VA-LRTI, * *p* =0.012 versus non-COPD patients with no VA-LRTI, *p* > 0.05 for all other comparisons between COPD and non COPD patients.

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
