# Peer review of "Impact of Chronic Obstructive Pulmonary Disease on Incidence, Microbiology and Outcome of Ventilator-Associated Lower Respiratory Tract Infections"

_microorganisms, 2020, doi:10.3390/microorganisms8020165_

Round 1

Reviewer 1 Report

In this resubmitted manuscript, Anahita Rouze et al made significant revision in the data quality and analysis. I have no further question.

Reviewer 2 Report

The reviced manuscript is well written. Thus, publish it in this form.

This manuscript is a resubmission of an earlier submission. The following is a list of the peer review reports and author responses from that submission.

Round 1

Reviewer 1 Report

The study determines the impact of COPD on incidence and outcomes of VA-LRTI (VAT and VAP) on 2960 TAVeM study subjects. The study has several interesting findings, some corroborating the literature, others not. Importantly, this study shows that COPD does not impact LRTI incidence, mortality, enrichment of MDR pathogens, duration of MV and length of ICU stay in VAP patients, etc. While this study was not initially designed to study COPD in this cohort, and therefore some important features such as severity of COPD, or long term oxygen therapy or non-invasive home ventilation history were not recorded in this study, however, it still represents one of the largest prospective multicenter study on an international cohort of VA-LRTI constituting both VAT and VAP; and therefore will be interesting for the field to take note of this study.

Reviewer 2 Report

In this manuscript, Anahita Rouze et al described their findings on the impact of COPD on ventilator associated lower respiratory tract infection. The design and data analysis is reasonable and convincing. The main concern is that the authors didn’t incorporate the specific analysis for different age and gender groups rather than roughly put all patients in either COPD or non-COPD into the cohort of analysis, which might be the main reason of non-significant result on mortality rate for COPD in VAP and VAT patients, as discussed by the authors. The subgroup analysis is recommended before further consideration. Also, as compared to other parts, the lack of insightful discussion in the bacteria and their contribution to COPD in mortality, extension of MV and antibiotic selection makes it flawed to be a paper for the journal of microorganism.